# Validation Testing of the European Portuguese Critical-Care Pain Observation Tool

**DOI:** 10.3390/healthcare10061075

**Published:** 2022-06-09

**Authors:** Rita Marques, Filipa Araújo, Marisa Fernandes, José Freitas, Maria Anjos Dixe, Céline Gélinas

**Affiliations:** 1Centre for Innovative Care and Health Technology (ciTechCare), Department of Nursing, Higher School of Health of The Portuguese Red Cross, 1300-125 Lisbon, Portugal; 2Department of Rehabilitation, University Hospital of Santa Maria, 1649-028 Lisbon, Portugal; lipa_penelope@msn.com; 3Intensive Care Unit, University Hospital of Santa Maria, 1649-028 Lisbon, Portugal; marisa.ac.fernandes@sapo.pt; 4Operating Room, University Hospital of Santa Maria, 1649-028 Lisbon, Portugal; jose_mpe0204@hotmail.com; 5Centre for Innovative Care and Health Technology (ciTechCare), Nursing School, Polytechnic of Leiria, 2411-901 Leiria, Portugal; maria.dixe@ipleiria.pt; 6Centre for Nursing Research, Ingram School of Nursing, Jewish General Hospital, McGill University, Montréal, QC H3T 1E2, Canada; celine.gelinas@mcgill.ca

**Keywords:** pain assessment, critical care nursing, validation

## Abstract

Aim. The study aim was to validate the Portuguese version of the Critical-Care Pain Observation Tool (CPOT) in the critically ill adult population of Portugal. Methods. A prospective, observational cohort study was conducted to evaluate the CPOT in mechanically ventilated patients who were admitted to an intensive care unit. A consecutive sample of 110 patients was observed at rest pre-procedure, during a nociceptive procedure (NP) which includes turning/positioning and endotracheal or tracheal suctioning and 20 min post-procedure. Two raters participated in the data collection. The discriminative validity, criterion validity, convergent validity and inter-rater reliability of the CPOT were examined. Results. The inter-rater reliability was excellent (0.93 ≤ α ≤ 1.00) at rest and fair to moderate (0.39 ≤ α ≤ 0.60) during the NP. The CPOT could discriminate between conditions with higher scores during the NP when compared to CPOT scores at rest (*p* < 0.001). The optimal CPOT cut-off score was >2, with a sensitivity of 71% and a specificity of 80%, and self-reported pain was the gold standard criterion. Significant correlations (<0.40) were found between CPOT scores, the heart rate and the respiratory rate during the nociceptive procedure. Conclusions. The CPOT appears to be a valid alternative for both ventilated and non-ventilated patients who are unable to communicate.

## 1. Introduction

Pain is a multidimensional and personal experience, and its assessment is key to optimal pain management [1]. It can cause pathophysiological changes, which can affect the patient’s level of consciousness, sleep, circulatory system, endocrine system, metabolism and/or gastrointestinal system, as well as psychological disorders [2,3,4,5]. Furthermore, as a major consequence, acute pain may evolve to chronic pain and can lead to various negative psychological outcomes [6].

The patient’s self-report should be obtained whenever possible, as it is the gold standard measure of pain. However, critically ill patients may be incapable of self-report due to mechanical ventilation and sedation [7,8]. Therefore, assessing pain in this population is a challenge for the intensive care unit (ICU) care team. In non-communicative patients, such as those who are mechanically ventilated (MV) and sedated, behavioural assessment tools (such as the Behavioural Pain Scal—BPS, and the Critical-Care Pain Observation Tool—CPOT) are alternative measures that can be used by the ICU care team to guide pain control interventions [1,2,7,8]. 

Considering their continuous presence at the bedside and their close proximity to the patient and his/her family, nurses play a key role in the assessment and management of pain [4]. Nurses can perform adequate pain assessment using validated tools, and regular pain monitoring is indispensable for optimal pain management. Indeed, the implementation of pain assessment tools has improved care outcomes, for example, by shortening the mechanical ventilation duration and ICU length of stay, reducing nosocomial infection rates and decreasing the number of long-term complications [6].

Studies have shown that the Behavioural Pain Scale (BPS) [9] and the Critical-Care Pain Observation Tool (CPOT) [10] are the most suitable alternative measures for pain assessment in critically ill patients unable to self-report.

However, in some studies, BPS scores increased during both painful and a nonpainful procedures, whereas CPOT scores only increased during a painful procedure [11,12]. Therefore, the CPOT was identified as the tool of choice for the assessment of pain in patients in the ICU with altered levels of consciousness [5,11,13]. The original BPS can only be used for mechanically ventilated patients, while for pain assessment in non-intubated patients, the BPS-non-intubated (NI) is the alternative option, and it is not yet available in Portuguese [14].

So far, the BPS has been the only validated pain scale in the Portuguese population and was selected by the Portuguese Society of Intensive Care [14]. Nevertheless, the Society of Critical Care Medicine suggests both the BPS/BPS-NI and the CPOT scales in their practice guidelines [4]. Therefore, their translation and validation are fundamental [15,16] to provide valid pain assessment tools in critically ill Portuguese patients.

This study aimed to translate the CPOT into Portuguese and validate the translated version. Our specific objectives were to determine the CPOT discriminative validity, criterion validity and convergent validity as well as the inter-examiner reliability of the Portuguese version.

## 2. Materials and Methods

### 2.1. Design

A prospective observational cohort study was conducted, reported according to the STROBE statement for observational studies [17].

### 2.2. Setting and Sample

We used a consecutive sample of 110 medical and surgical patients who were admitted to a 11-bed intensive care unit (ICU), at a University Hospital located in Lisbon (Portugal). For this study, we defined the inclusion criteria described below: (a) were admitted to the ICU; (b) had a minimum length of stay in the ICU of 24 h; (c) were 18 years old or older; (d) were mechanically ventilated; and (e) were able to understand Portuguese before intubation. Patients with neurological deficits (such as reduced range of movement, decreased strength and functionality, altered sensitivity) were excluded. 

### 2.3. Data Collection Tools and Methods

In the initial stage of the study, following a written authorisation by the author of the CPOT (CG) and after obtaining permission to use the English version of the CPOT, the tool was translated into Portuguese. International guidelines relevant to the cross-cultural adaptation process were followed to ensure linguistic equivalence, conceptual equivalence and psychometric equivalence [18,19].

The process included the following steps: preparation, forward-translation, reconciliation, back-translation, back-translation review, expert panel and pilot testing.

The translation from English to Portuguese was carried out by two professional translators. After the translations were analysed and the consensus version was obtained, two professional English translators performed the back-translation. Following the results for the back-translation, all versions were harmonised to detect and address any discrepancies that might arise between different language versions, ensuring conceptual equivalence. 

Content and face validity were submitted to a Delphi panel consisting of eight nurses (with a master’s degree and experience working in the ICU) and two PhD experts in scale validation. Each of the Delphi panel members gave their opinion on the adequacy of the translation and the relevance of the item. The level of agreement for each item was 100%. The Portuguese translation was pilot tested by three ICU nurses on 15 patients to check the interpretation and ease of comprehension. No changes were necessary. 

Following the completion of the transcultural adaptation process, we appraised the CPOT’s psychometric characteristics. The patients were assessed with the CPOT and the BPS (the BPS was used for convergent validity) at rest pre-procedure, during a nociceptive procedure (NP) and 20 min after the procedure (t2), for a total of three assessments of each patient. The NP included one of the following standard care procedures commonly used in previous validation studies: turning/positioning and endotracheal or tracheal suctioning [20,21,22,23,24]. Pain assessments were performed simultaneously and independently by two trained ICU nurses during the daytime. 

The raters (ICU nurses) received a 90-min standardised training session (theoretical and practical) from the primary investigator, during which they were taught how to use the CPOT and practiced its administration, inspired by the one developed by the authors of the CPOT [11,24]. Following training, they completed bedside assessments with the CPOT in the presence of the study investigator to ensure that the tool was used appropriately. 

The CPOT and the BPS were used for all participants. In several studies, authors reported a correlation coefficient between the scores of two scales (e.g., BPS and CPOT) and described this strategy as convergent or criterion validation [20]. 

Whenever the BPS score exceeded 5, pain was considered to be present. The inter-rater reliability of nurses’ CPOT scores was described at each time point. Comparison of CPOT scores at rest pre- and post-procedure with the NP allowed the examination of discriminative validity. Criterion validity was established using the BPS threshold for the absence (BPS ≤ 5) versus the presence (BPS > 5) of pain to examine their association with CPOT scores. The association between the two scales (BPS and CPOT) was commonly performed as a validation strategy in previous studies [20]. Finally, associations between vital signs (i.e., heart rate [HR], mean arterial pressure [MAP] and respiratory rate [RR]) extracted from the bedside monitors, as well as CPOT scores, were described at each time point for the examination of convergent validity. 

### 2.4. Instruments

Critical-Care Pain Observation Tool: The CPOT was initially developed in French Canadian by Gélinas and colleagues in 2006 based on a thorough content validation process and was translated into English using a forward–backward method [20]. The CPOT is divided into four sections, each pertaining to different behavioural categories: facial expression, body movements, muscle tension and compliance with the ventilator (only for mechanically ventilated patients) or vocalisation (only for non-intubated patients). Each section is scored from 0 to 2, and the possible total score ranges from 0 to 8 [10]. The CPOT is available in at least 19 languages and has been tested in almost 4000 ICU patients [20]. The initial validation testing was achieved using a convenience sample of 105 patients. All participants were evaluated when mechanically ventilated (64 unconscious and 51 conscious), as well as post-extubation. The validation findings supported inter-rater reliability, discriminative validity between nociceptive and non-nociceptive procedures, criterion validity with positive associations between CPOT scores and self-reported pain scores [10]. It was found to be feasible and clinically relevant by ICU nurses [20]. 

The raters completed a 90-min standardised training session, inspired by the one developed by the author of the CPOT, during which they were taught how to use the CPOT and practiced scoring using the tool in the presence of the primary investigator [11,24]. Following training, they completed bedside assessments using the CPOT in the presence of the study investigator to ensure that the tool was used appropriately. 

Behavioural Pain Scale: The BPS was initially developed in French from France by Payen and colleagues in 2001 and includes four behavioural indicators: facial expression, upper limb movements and compliance with mechanical ventilation. Each item is scored from 1 to 4, and the possible total score ranges from 3 to 12 [9]. Its initial validation testing in a sample of 30 mechanically ventilated ICU patients (269 observations) showed good inter-rater reliability and discriminative validity between nociceptive and non-nociceptive procedures. The Brazilian version of the BPS showed good agreement between raters (intraclass correlation coefficient (ICC) = 0.80 and 0.97) and good criterion validity with self-reported pain scores. Discriminative validity led to variable findings across studies, including significant increases during both nociceptive and non-nociceptive procedures [25].

Socio-demographic and clinical information: Sociodemographic (sex and age) and clinical data (medical or surgical diagnosis, Glasgow Coma Scale (GCS), presence of continuous and/or intermittent sedation and analgesia), as well as vital signs (MAP, HR and RR) available through continuous monitoring in the ICU, were also collected.

The GCS is used to assess the patient’s level of consciousness, which includes three sections: eye opening (scored from 1 to 4), verbal response (scored from 1 to 5) and motor response (scored from 1 to 6). The probable score ranges from 3 (unconscious, the worst response) to 15 (fully conscious and oriented, the best response). Patients were classified as unconscious with GCS ≤ 8 and conscious with GCS > 9.

### 2.5. Data Analysis

Data were analysed using SPSS version 21 for Windows (IBM Corp., Armonk, NY, USA). Descriptive data analysis was performed using frequencies, percentages, mean  ±  standard deviation (SD) and ranges. Data normality was analysed through the Shapiro–Wilk test. Several non-parametric tests were used to assess validity, and the alpha was set at 0.05. The Wilcoxon Signed-Rank test (Z) was performed to evaluate the differences in CPOT scores at all time points (discriminative validity). The Mann–Whitney test was performed to compare CPOT scores using the BPS threshold for the absence versus the presence of pain as the reference standard, and receiver operating characteristic (ROC) curve analyses were also performed (criterion validity). ROC analyses is a common strategy to establish the performance of a tool to detect a condition (e.g., pain) as well as the sensitivity and specificity associated with the best threshold [20]. Spearman correlations between CPOT scores and vital signs (MAP, HR, RR) were obtained (convergent validity). Finally, inter-rater reliability between ICU nurses’ CPOT scores was estimated using weighted kappa coefficients (95% CI) at each time point. Values from 0.40 to 0.60 were considered moderate, and values above 0.60 were considered excellent [26].

### 2.6. Ethical and Institutional Approvals

The study was conducted in accordance with the Declaration of Helsinki [27], and the protocol was approved by the Ethics Committee (no: 423/16). Trained ICU nurses explained the study to eligible patients or their relatives, and informed written consent was obtained. When the patient was not able to provide his/her informed written consent, a close relative/legal representative was asked to complete the written consent form on the patient’s behalf. Personal data were processed in accordance with the EU General Data Protection Regulation (GDPR2016/679). 

## 3. Results

Sample Description: The sample included 110 participants. A total of 38.2% of them were women, and their ages ranged from 20 to 95 (67.24 ± 14.61) years. Almost half of the participants were admitted to the ICU for a surgical diagnosis (56, 51%), 53 did not receive analgesia (48%) and 93 did not receive any sedation (85%). Patients were classified as ‘unconscious’ (63, 57%) or ‘conscious’ (47, 43%). 

Validity: Discriminative validity was supported by higher median CPOT scores during NP, when compared to CPOT scores at rest pre-procedure (baseline). Within-group analysis revealed higher CPOT scores during NP compared with those obtained at rest (*p* < 0.001) (Table 1).

The ROC of Rater 1 CPOT scores during the nociceptive procedure for the total sample is presented in Figure 1.

The ROC of Rater 1′s scores during NP showed good performance of the CPOT to classify patients with pain based on BPS > 5 (AUC = 0.764; SE = 0.077; 95% CI 0.613–0.915; *p* < 0.001) with a sensitivity of 71% and a specificity of 80%. A similar performance of the CPOT was found based on ROC analysis of Rater 2’s scores. In this sample, the optimal CPOT threshold was ≥3.

According to the convergent validity, associations between CPOT scores and vital signs were evaluated; therefore, we present below the mean and standard deviation values of the vital signs: mean arterial pressure (MAP), heart rate (HR) and respiratory rate (RR), at each time point. At rest pre-procedure: MAP = 84.50 (16.47), RR = 21.50 (6.14) and HR = 85.10 (19.45); during a nociceptive procedure: MAP = 94.14 (16.40), RR = 23.97 (6.48) and HR = 91.10 (20.53); 20 min after the procedure: MAP = 83.28 (17.34), RR = 20.68 (6.02) and HR = 85.97 (19.16).

Positive mild associations were found between CPOT scores, HR (both raters) and RR (Rater 2) (Table 2). Very low to no associations were obtained between CPOT and MAP.

Regarding criterion validity, CPOT scores during NP were significantly associated with the presence of pain (BPS > 5) for Rater 1 and Rater 2 (Table 3). In fact, CPOT scores were higher in patients with BPS > 5 (presence of pain) when compared to those with BPS scores ≤ 5 (absence of pain). 

Inter-rater reliability: Weighted kappa coefficients between ICU nurses’ CPOT scores ranged from 0.39 to 1.00, indicating moderate to excellent inter-rater reliability.

## 4. Discussion

This study aimed to translate and validate the Portuguese version of the CPOT in 110 medical and surgical mechanically ventilated ICU patients, either conscious or unconscious. The level of consciousness and sedation may influence pain behaviours, which are less frequent or blurred in patients who are unconscious or heavily sedated [20]. 

In this validation process, we examined the discriminative validity, criterion validity, convergent validity and inter-rater reliability of the CPOT. Through these validation strategies, we were able to determine the tool’s psychometric characteristics in this Portuguese sample.

Similar to previous studies [3,4,10,11,13,21,28,29], scores on the Portuguese version of the CPOT increased during common standard care procedures compared to rest in both conscious and unconscious groups, demonstrating discriminative validity. Criterion validity was also demonstrated, with significant associations of CPOT with the BPS threshold as the reference standard. During NP, higher CPOT scores were obtained in patients with BPS > 5 compared to those with BPS ≤ 5. ROC findings showed a moderate performance (AUC 0.6–0.8) [28] of the CPOT to detect pain during NP with a threshold of 3, as found in previous studies [20,29,30]. In our study, correlations between vital signs and CPOT scores were low to very low; therefore, they did not support convergent validity and lacked clinical relevance to pain assessment. Vital signs were not found to be good indicators for pain assessment in the ICU [11,26,28,31] when considered individually [2,20,32], because they are also influenced by other factors (such as sedation, anxiety, difficulty breathing or sepsis) [13].

Inter-rater reliability was satisfactory. Most kappa coefficients were moderate to excellent [26]. These results demonstrated that both ICU nurses obtained consistent CPOT scores at each time point following standardised training. Similar findings were reported in previous studies [33].

In this study, we found that the correlation between vital signs and the score obtained with the CPOT assessment were low, because when individually assessed vital signs were not relevant, they could have been influenced by other factors such as anxiety, sedation or breathing difficulties.

The Danish study [30] confirms that pain cannot be evaluated only through physiological indicators, even when the patient is awake, without sedation due to the changes that they may be experiencing due to the patient’s clinical situation (sepsis anxiety).

Our study, as well as the Swedish [16] and the Danish [30], was carried out by two independent evaluators and proved that CPOT should be applied to critically ill patients who are unable to verbalise, showing good inter-rater reliability.

As in previous studies on this study, 53 patients did not receive analgesia and 93 patients did not receive sedation, which allowed for the observation of higher CPOT scores during painful procedures in comparison to non-painful procedures. However, it was not possible to verify a correlation between pain and PFM increase, which justifies the aforementioned statements.

In our study, we used another BPS behavioural assessment scale as a comparison in our pain assessment, which showed a good correlation between the two. In the Danish study [30], the CAM-ICU scale was applied concomitantly, in which 28 patients had a positive CAM ICU, which may affect the reliability of pain self-assessment, which is considered the gold standard.

Despite this, we can say that the results obtained in the different studies were convergent, with regard to the increase in the CPOT score during painful procedures.

### 4.1. Limitations

Some limitations must be mentioned. Due to difficulties in reconciling the presence of three raters in all instants of assessment, we chose to use only two. This strategy had already been employed in other validation studies [16,30]. However, some authors suggest that, to evaluate the CPOT’s inter-rater reliability, more than two raters should be involved in the process [11,20].

### 4.2. Implications and Recommendations for Practice

The Portuguese version of the CPOT was shown to be valid in this sample of Portuguese mechanically ventilated ICU patients, who were conscious or unconscious. Portuguese ICU nurses should not rely on vital signs for pain assessment and are encouraged to use a valid behavioural scale such as the CPOT when they suspect that their patient may be in pain.

## 5. Conclusions

The Portuguese version of the CPOT seems to be a valid and reliable tool for pain assessment in mechanically ventilated ICU patients, whether they are conscious or unconscious. Thus, the CPOT is an alternative option to the BPS which, until now, has been the sole validated scale for pain assessment in Portuguese ICU patients. The CPOT can be applied to ICU patients who are incapable of communicating verbally or using signs, whether they are mechanically ventilated or not.

The inter-rater reliability of the CPOT was excellent at rest and fair to moderate during the nociceptive procedure. The CPOT could discriminate between conditions with higher scores during the nociceptive procedure when compared to CPOT scores at rest. The optimal CPOT cut-off score was >2 with a sensitivity of 71% and a specificity of 80% using self-reported pain as the gold standard criterion. Significant but low correlations were found between CPOT scores, the heart rate and the respiratory rate during the nociceptive procedure.

Nurses have a fundamental role to play in monitoring and managing pain. For this purpose, they must use appropriate pain assessment tools adapted to the patient’s ability to communicate their pain, such as self-report or behavioural scales. In Portugal and in addition to the BPS, the CPOT appears to be another valid alternative scale to use because it is applicable to patients unable to self-report, whether they are mechanically ventilated or not.

## Figures and Tables

**Figure 1 healthcare-10-01075-f001:**
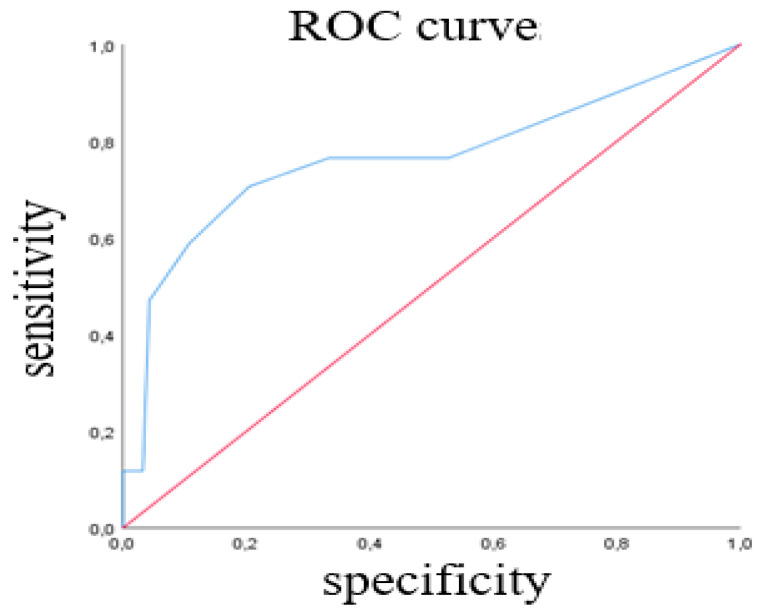
ROC curves for the CPOT cut-off.

**Table 1 healthcare-10-01075-t001:** Critical-Care Pain Observation Tool scores recorded at each time point by Rater 1 and Rater 2 in unconscious and conscious mechanically ventilated patients.

		Median	IQR ^a^	Z	*p*
Unconscious endotracheal intubated patients (*n* = 63)					
Rater 1	At rest, baseline (t0)	0	1 (1-0)	−6.005 ^b^	0.000 ^d^
	During the nociceptive procedure (t1)	2	3 (4-1)		
Rater 2	At rest, baseline (t0)	0	1 (1-0)	−3.461 ^b^	0.001 ^c^
	During the nociceptive procedure (t1)	0	1 (1-0)		
Rater 1	During the nociceptive procedure (t1)	2	3 (4-1)	−6.005 ^b^	0.000 ^d^
	20 min after the procedure (t2)	0	0 (0-1)		
Rater 2	During the nociceptive procedure (t1)	0	1 (1-0)	−3.985 ^b^	0.000 ^d^
	20 min after the procedure (t2)	0	0 (0-1)		
Conscious endotracheal intubated patients (*n* = 47)					
Rater 1	At rest, baseline (t0)	0	0 (0-0)	−5.816 ^b^	0.000 d
	During the nociceptive procedure (t1)	3	3 (5-2)		
Rater 2	At rest, baseline (t0)	0	0 (0-0)	−4.752 ^b^	0.000 ^d^
	During the nociceptive procedure (t1)	2	4 (4-0)		
Rater 1	During the nociceptive procedure (t1)	3	3 (5-2)	−5.747 ^b^	0.000 ^d^
	20 min after the procedure (t2)	0	0 (0-0)		
Rater 2	During the nociceptive procedure (t1)	2	4 (4-0)	−4.672 ^b^	0.000 ^d^
	20 min after the procedure (t2)	0	0 (0-0)		

^a^ Interquartile range; ^b^ Based on negative ranks; ^c^
*p* < 0.01; ^d^
*p* < 0.001.

**Table 2 healthcare-10-01075-t002:** Spearman correlations between Critical-Care Pain Observation Tool scores and vital signs for Rater 1 and Rater 2 at each time point.

	Time Point	MAP	HR	RR
Rater 1	At rest, baseline	0.001	0.267 ^a^	−0.034
	During the nociceptive procedure	0.069	0.272 ^a^	0.162
	20 min after the procedure	−0.100	0.272 ^a^	−0.011
Rater 2	At rest, baseline	−0.020	0.179	0.015
	During the nociceptive procedure	0.129	0.372 ^a^	0.323 ^a^
	20 min after the procedure	−0.036	0.310 ^a^	0.031

^a^ Correlation is considered most significant at the 0.01 level; MAP—mean arterial pressure; HR—heart rate; RR—respiratory rate.

**Table 3 healthcare-10-01075-t003:** CPOT scores in patients with and without pain according to the Behavioural Pain Scale threshold for Rater 1 and Rater 2 at each time point.

	Time Point	BPS Threshold ^a^	CPOT					
			Quartile 25	Quartile 50	Quartile 75	Mean Rank	U	*p*
Rater 1	t0	Yes, present No, absent	0 0	0 0	2 0	64.1553.21	801.500	0.049 ^b^
	t1	Yes, present No, absent	3 1	5 3	6 4	77.9550.81	438.000	0.001 ^c^
	t2	Yes, present No, absent	0 0	0 0	2.5 1	62.0055.13	273.000	0.511
Rater 2	t0	Yes, present No, absent	0 0	0 0	1 0.5	54.0055.57	255.000	0.886
	t1	Yes, present No, absent	1 0	5 1	6 2	80.0651.01	373.000	0.000 ^d^
	t2	Yes, present No, absent	0 0	0 0	2.5 0	63.0055.07	267.000	0.432

t0 at rest-baseline; t1 during the NP; t2 20 min after the procedure; ^a^ Without pain (BPS ≤ 5) or with pain (BPS > 5); ^b^
*p* < 0.05; ^c^
*p* < 0.01; ^d^
*p* < 0.001.

## Data Availability

All data are available from the corresponding author upon reasonable request.

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
