# Peer review of "Validation Testing of the European Portuguese Critical-Care Pain Observation Tool"

_healthcare, 2022, doi:10.3390/healthcare10061075_

Round 1

Reviewer 1 Report

I have read your manuscript with great interest. Pain management in the ICU patient is a research field not yet fully explored.I like your study but some ethical perplexities. In fact, regarding the consent, you say that it is the legal representatives who signed the consent.How were the legal representatives identified? Did the Ethics Committee give you precise instructions regarding informed consent? Wouldn't it have been better to select patients from those candidates for intensive care after an election procedure?In this way, patients would have signed the consent without misunderstanding. I ask you to answer my questions to better clarify my doubt. otherwise an excellent manuscript.

Kind Regards

Author Response

We include

In this study, we found that the correlation between vital signs and the score obtained with the CPOT assessment were low, because when individually assessed vital signs were not relevant because they could have been influenced by other factors such as anxiety, sedation or breathing difficulties.

The Danish study[30] confirms that pain cannot be evaluated only through physiological indicators, even when the patient is awake, without sedation due to the changes that they may be experiencing due to the patient's clinical situation (sepsis anxiety).

Our study, as well as the Swedish[16] and the Danish[30], was carried out by two independent evaluators and proved that CPOT should be applied to critically ill patients who are unable to verbalize, showing good inter-rater reliability.

As in previous studies on this study, 53 patients did not receive analgesia and 93 patients did not receive sedation, which allowed to observe higher CPOT scores during painful procedures in comparasion to non-painful procedures. However, it was not possible to verify a correlation between pain and PFM increase, which justifies the aforementioned statements.

In our study, we used another BPS behavioural assessment scale as a comparison in our pain assessment, which showed a good correlation between the two. In the Danish study[30], the CAM-ICU scale was applied concomitantly, in which 28 patients had a positive CAM ICU, which may affect the reliability of pain self-assessment, which is considered Gold Standard.

Despite this, we can say that the results obtained in the different studies were convergent, with regard to the increase in the CPOT score during painful procedures

Reviewer 2 Report

In the abstract my suggestion is to say which nociceptive procedures were performed. I have no other comment or suggestion.

Congratulation for the study.

Author Response

In the abstract my suggestion is to say which nociceptive procedures were performed. I have no other comment or suggestion. 

we include in the abstract  “which includes turning/positioning and endotracheal or tracheal suctioning “. because of this inclusion the summary has 206 words  

Reviewer 3 Report

Marques et al have presented an excellent research paper that describes the validation process for the Portuguese translation of CPOT in critically ill patients. They have justified and explained their study objectives, described their methodology in sufficient detail, and presented the results in a concise manner. 

I would additionally like to see an expansion of the Discussion section. In particular, how do the results of this validation study compare with other studies that have attempted validation of translations into other languages? For example, I see two references in the reference list (reference 16 and 30) that studied the validation of Swedish and Danish translation of COPT. Can the authors elaborate how their results compare with the Danish and Swedish studies? If the results are different, then what would explain the difference?

Author Response

The ICU where the study was carried out receives patients in an acute critical situation, after surgical treatment or trauma victims, so it is not possible to ask for free and informed consent. Consent was requested from the reference person/ legal representative, an element who receives information about the patient's clinical history and collaborates with the health team in decision-making about their disease situation.

We include close relative/legal representative